# Healthy food retail availability and cardiovascular mortality in the United States: a cohort study

Gina S Lovasi ,[1] Norman J Johnson,[2] Sean F Altekruse,[3] Jana A Hirsch,[1] Kari A Moore,[1] Janene R Brown ,[1] Andrew G Rundle,[4] James W Quinn,[4] Kathryn Neckerman,[4] David S Siscovick[5]

[1]Urban Health Collaborative, Dornsife School of Public Health, Drexel University, Philadelphia, Pennsylvania, USA
[2]Center for Administrative Records and Research Applications, Census Bureau, Washington, District of Columbia, USA
[3]National Heart Lung and Blood institute, Division of Cardiovascular Sciences, National Institutes of Health, Bethesda, Maryland, USA
[4]Built Environment and Health Research Group, Columbia University, New York, New York, USA
[5]New York Academy of Medicine, New York, New York, USA

**Correspondence to**
Dr Gina S Lovasi;
gsl45@drexel.edu

## ABSTRACT

**Objectives** We investigated the association of healthy food retail presence and cardiovascular mortality, controlling for sociodemographic characteristics. This association could inform efforts to preserve or increase local supermarkets or produce market availability.

**Design** Cohort study, combining Mortality Disparities in American Communities (individual-level data from 2008 American Community Survey linked to National Death Index records from 2008 to 2015) and retail establishment data.

**Setting** Across the continental US area-based sociodemographic and retail characteristics were linked to residential location by ZIP code tabulation area (ZCTA). Sensitivity analyses used census tracts instead, restricted to urbanicity or county-based strata, or accounted for non-independence using frailty models.

**Participants** 2 753 000 individuals age 25+ living in households with full kitchen facilities, excluding group quarters.

**Primary and secondary outcome measures** Cardiovascular mortality (primary) and all-cause mortality (secondary).

**Results** 82% had healthy food retail (supermarket, produce market) within their ZCTA. Density of such retail was correlated with density of unhealthy food sources (eg, fast food, convenience store). Healthy food retail presence was not associated with reduced cardiovascular (HR: 1.03; 95% CI 1.00 to 1.07) or all-cause mortality (HR: 1.05; 95% CI 1.04 to 1.06) in fully adjusted models (with adjustment for gender, age, marital status, nativity, Black race, Hispanic ethnicity, educational attainment, income, median household income, population density, walkable destination density). The null finding for cardiovascular mortality was consistent across adjustment strategies including minimally adjusted models (individual demographics only), sensitivity analyses related to setting, and across gender or household type strata. However, unhealthy food retail presence was associated with elevated all-cause mortality (HR: 1.15; 95% CI 1.11 to 1.20).

**Conclusions** In this study using food establishment locations within administrative areas across the USA, the hypothesised association of healthy food retail availability with reduced cardiovascular mortality was not supported; an association of unhealthy food retail presence with higher mortality was not specific to cardiovascular causes.

## Strengths and limitations of this study

► In light of the ongoing salience of 'food deserts' in policy discussions, separate consideration of healthy food store presence while controlling for potential socioeconomic confounders may reveal whether policy strategies with a focus on preserving or increasing healthy food retail are likely to improve cardiovascular outcomes.

► Data are from the Mortality Disparities in American Communities (MDAC) study, a large US-based representative sample that combines the strengths of the American Communities Survey, individual linkage to the National Death Index, and area-based characteristics.

► Our approach assessed the robustness of findings across adjustment strategies, population strata (women, men, urban residents, single-family households and county-based groupings), analytical approaches, geographic units (postal codes or census tracts) and with variation in exposure and outcome definitions.

► Key limitations include the risks of uncontrolled confounding, exposure misclassification, incomplete outcome ascertainment, and selection bias.

## INTRODUCTION

Modifiable risk factors are associated with more than 70% of clinical cardiovascular disease (CVD),[1] the leading cause of death in the USA.[2] Built environment characteristics may affect health-related behaviours that contribute to chronic disease risk, including cardiovascular morbidity and mortality,[1] potentially explaining geospatial variation in cardiovascular outcomes.[3–6]

The built environment could be improved as a component of population-level CVD prevention efforts. Concepts such as food deserts have particular resonance in policy discussions.[7] Studies typically define food deserts through both low-income criteria and a lack of healthy food retail, as in a recent example.[8] Scarcity of healthy food retail

may hinder individuals' and families' efforts to eat nutritious diets that include fresh foods.[9–13] Yet healthy food availability depends on neighbourhood socioeconomic context.[10–12] An operationalisation of food deserts that conflates inadequate access to healthy food retail and low area-based income can provide evidence for a policy approach that jointly tackles these challenges. However, separate consideration of healthy food store availability may better address the likely health implications of policy strategies with an exclusive focus on preserving or increasing healthy food retail.[14]

In the present study, we use food retail data linked to the Mortality Disparities in American Communities (MDAC) study. Individual and household socioeconomic data and food retail data[15] are from the 2008 American Community Survey (ACS), with outcome assessment based on National Death Index (NDI) linkage. Our analytic approach uses survival analyses, minimally adjusted for demographic characteristics, considering further adjustment for socioeconomic and contextual characteristics. We hypothesised that presence of healthy food sources in the home postal code area, operationalised using ZIP code tabulation areas (ZCTAs), would be associated with lower cardiovascular mortality. We consider whether food environment-mortality associations were consistent across population strata, alternative exposure and outcome specifications, and analytic approaches.

## METHODS
### Study sample and data linkage overview
Individual linkage of data from 2008 ACS respondents to the NDI provides a foundation for MDAC, a collaborative project of the US Census Bureau, the Centers for Disease Control and Prevention and the National Institutes of Health.[16] The ACS sampling frame is designed to be representative across demographic categories (age, sex, race, ethnicity and state of residence) for the US population. Sampling weights are based on annual ACS national population estimates from the US Census Bureau.

Geographic linkage used residential ZCTA and census tract. Intending to capture food environment retail reachable within a short drive,[17] ZCTA was selected as the primary level for contextual characteristics during the MDAC proposal approval process, with a planned sensitivity analysis using census tract data. Both ZCTA and census tract geographies are systematically larger in areas of low population density.

### Patient and public involvement
The analyses presented in this manuscript were investigator-initiated and did not reflect patient or public involvement, though such involvement shows promise to provide a foundation for the innovation and relevance of future inquiry.

### Inclusion criteria
Our analytic sample was initially restricted to individuals from ACS survey households with consent for research data use (N=4 512 000; note that sample sizes in tables and to illustrate changes as inclusion criteria are applied are rounded to the thousands during disclosure proofing; CBDRB-FY20-CES004-021). We further limited to individuals for whom personal identifiers were sufficiently complete to allow linkage to NDI through 31 December 2015 (4 480 000). Due to potential differences in food acquisition, we excluded individuals residing in group quarters or in households without a full kitchen (3.8%). Linkage to ZCTA-level food environment data assembled across the continental US was completed for 4 107 000 individuals. Based on our interest in associations with cardiovascular mortality adjusted for individual socioeconomic characteristics, we restricted our analyses to adults 25+ years of age (2 923 000). Final exclusion of observations with missing covariate data resulted in an analytic sample of 2 753 000.

### Geographic units and their characterisation
Contextual characteristics were assembled and linked to geocoded home address data using ZCTA and census tract boundaries (TIGER Line, 2016 version of the 2010 census boundaries). The area-based characteristics considered as potential confounders, including population density and median household income, used ACS-based estimates for the years 2008 to 2012 included in a harmonised Longitudinal Tract Database.[18]

Food retail characteristics were estimated using National Establishment Time Series (NETS) data. Steps to enhance accuracy, consistency, and replicability of our work with these data have been described elsewhere, along with the rationale and checking of our business category definitions.[15]

A combined category of healthy food retail sources was defined to include supermarkets (using chain name searches, 8-digit Standard Industrial Codes (SIC), and size thresholds: number of employees ≥25 or sales volume ≥ $2 million) and produce stores (fruit and vegetable market SIC codes). A secondary definition of healthy food sources included additional retail that may provide some cardioprotective benefits, but which are less common and have received limited attention in the literature (natural food, health food and vitamin stores). For unhealthy food retail, we considered a combined category of fast food, quick service, and pizza restaurants; bakery, ice cream, coffee, and candy shops; and convenience and small grocery stores. A broadened definition of unhealthy food retail sources further included as potential sources of highly processed foods: pharmacies, gas stations, and nut stores (typically selling sweetened nuts and candy). While we recognise that establishments within the above categories offer items with varying nutritional value, our categorisation was informed by prior literature and by the relative affordability of and salience of fresh items.

In addition to food retail, our maximally adjusted models control for a broader retail category labelled 'walkable destinations' designed to include establishments that contribute to making pedestrian transportation attractive and feasible.[19]

We operationalised these retail categories across 1990–2014 NETS data, which contained approximately 58 million unique establishments identified by DUNS number (establishments had a mean of 1.3 distinct addresses reported over time, yielding more than 77 million records to re-geocode).[15] For alignment with MDAC baseline, we use retail data from 2008 across 32 170 ZCTAs and 72 246 census tracts. Count of establishments was constructed for each retail category, dichotomised as present/absent, and used to estimate density using a land area denominator (count/km$^2$).

### Individual demographic and household socioeconomic data

Demographic characteristics from the ACS included gender, age, marital status, nativity (US born vs other), and race/ethnicity. Socioeconomic characteristics included educational attainment, and household income. To increase interpretability, age was rescaled to 10-year increments, and income was rescaled to increments of US$10 000.

### Defining urban and county-based strata

Residential location of each MDAC household was classified as urban if located within an urbanised area (UAs) or urban cluster (UCs). UAs consist of densely developed territories that contain 50 000 or more people. UCs consist of densely developed territories with at least 2500 people but fewer than 50 000 people. In 2010, an estimated 81% of the US population resided in urban areas.[20]

A county-level analysis inspired by prior work on the 'Eight Americas'[21] was conducted by Jahn Hakes and Sean Altekruse (personal communication, 2 June 2020), resulting in 11 strata across the continental USA (additional strata defined for Alaska and Hawaii are not used here). Briefly, 39 county-level sociodemographic and climate variables (sourced from ACS and CDC WONDER[22]) were used in a principle component analysis, resulting in six components that were then used to assign counties into strata with ad hoc names (Southern Rural, North Central, Mid-Sized, Sunbelt, Poor, Mountain West, Beach, Wealthy, Middle, Northern Tier and Big City America).

### All-cause and CVDs mortality outcome definitions

The primary cardiovascular mortality outcome based on NDI (based on 113 selected causes of death as defined by the Center for Disease Control and Prevention National Center for Health Statistics) included acute myocardial infarction, other acute ischaemic heart diseases, atherosclerotic CVD, atherosclerosis and all other forms of chronic ischaemic heart disease. As an alternative cardiovascular mortality outcome, we considered a broadened cardiometabolic mortality outcome category that includes causes of death noted above plus those related to diabetes mellitus, hypertensive heart disease, hypertensive heart and renal disease, heart failure, all other forms of heart disease, essential (primary) hypertension and hypertensive renal disease, cerebrovascular diseases, aortic aneurysm and dissection, other diseases of arteries, arterioles and capillaries, and other disorders of circulatory system. All-cause mortality was considered as a secondary outcome, used to evaluate the specificity of any associations with cause-specific mortality.

### Statistical analyses

Cox proportional hazards models used as an origin the date of ACS survey response, and end of follow-up was the date of death or 31 December 2015. The proportional hazards assumption for our exposure of interest was tested, with no significant violation detected (for the minimally adjusted model p=0.45, for the moderately adjusted model p=0.72, and for the fully adjusted model p=0.91; CBDRB-FY21-CES004-020). For cause-specific mortality analyses, death from other causes was treated as censoring. Non-independence across geographic units was accommodated through complex stratified random sample and corresponding weighting. In a sensitivity analysis, we considered frailty models accounting for clustering by county as an alternative modelling strategy.[23]

Indicators of healthy or unhealthy food retail presence were dichotomised and considered separately (not mutually adjusted due to multicollinearity concerns, based on individual-level Spearman's correlation coefficients among continuous contextual characteristics). All models minimally adjusted for demographic characteristics (age, marital status, nativity, race and ethnicity). Additional adjustment was added for educational attainment and household income, and then for contextual characteristics (area-based income, population density, and walkable destination density), both overall and for stratified analyses.

Analyses were conducted in SAS V.9.4, with data storage and access restricted to devices at Census Headquarters in Suitland, Maryland, USA; remote access for viewing output was provided through the Research Output Direct Access System system, available to GL and JRB following completion of requirements for Special Sworn Status.

## RESULTS

Of 2 753 000 individuals age 25+ living in households with full kitchen facilities, 82% had healthy food retail (supermarket or produce market) within their ZCTA (table 1). Those without healthy food retail were more likely to be married, born in the USA, White and Non-Hispanic. Those with healthy food retail had higher educational attainment and household incomes, and lived in areas with higher income, population density, walkable destination density and unhealthy food source density.

Density of retail establishments posited to be healthy (whether defined as supermarkets alone,

**Table 1** Demographic, socioeconomic and contextual characteristics among included Mortality Disparities in American Communities participants by availability of healthy food retail in residential ZIP code tabulation areas

| | No supermarket or produce market (N=492 000*) | Any supermarket or produce market (N=2 261 000*) | Total (N=2 753 000*) |
|---|---|---|---|
| **Individual demographic characteristics** | | | |
| Gender, % female | 52.0% | 53.3% | 53.1% |
| Age, mean (SD) | 52.8 (15.7) | 51.5 (16.0) | 51.8 (16.0) |
| Marital status, % married | 69.6% | 63.9% | 64.9% |
| Nativity, % US born | 95.4% | 85.6% | 87.3% |
| Race/ethnicity, % Black | 4.6% | 9.5% | 8.6% |
| Race/ethnicity, % White | 92.0% | 84.9% | 85.5% |
| Race/ethnicity, % Hispanic | 4.1% | 10.6% | 9.4% |
| Race/ethnicity, % Asian/Pacific Islander | 1.3% | 4.6% | 4.0% |
| Race/ethnicity, % other | 2.1% | 1.8% | 1.9% |
| **Socioeconomic characteristics** | | | |
| Educational attainment, % college or more | 21.9% | 31.0% | 29.3% |
| Annual income in $ US, mean (SD) | 71 800 (76 600) | 84 700 (95 300) | 82 400 (92 400) |
| **Contextual (ZCTA-based)** | | | |
| Median household income, mean (SD) | 55 300 (19 200) | 59 800 (22 800) | 59 000 (22 300) |
| Population density (thousands of residents/km$^2$), mean (SD) | 24 (83) | 144 (355) | 123 (327) |
| Walkable destination density (count/km$^2$), mean (SD) | 0.5 (3.0) | 3.1 (10.0) | 2.6 (9.2) |
| Fast food density | 0.2 (1.0) | 0.7 (1.8) | 0.6 (1.7) |
| Unhealthy food sources, restricted | 0.5 (2.8) | 3.1 (9.7) | 2.6 (8.9) |
| Unhealthy food sources, unrestricted | 0.5 (3.2) | 3.7 (11.2) | 3.2 (10.3) |

*Exact sample size suppressed during disclosure proofing; CBDRB-FY20-022.
ZCTA, ZIP code tabulation area.

supermarkets and produce markets, or a more inclusive definition including natural, health and vitamin stores) was correlated with unhealthy sources (person-level Spearman's correlation coefficients from 0.85 to 0.94). Strong correlations were also noted between food environment densities and both population density and walkable destination density (table 2).

Presence of healthy food within the ZCTA was not associated with reduced cardiovascular mortality across adjustment strategies considered (table 3). Similar patterns were observed in analyses that were sex stratified, restricted to urban residents, or restricted to households without multiple subfamilies (online supplemental figure S1, online supplemental tables S1–S4). Conditional associations accounting for random effects by county using frailty models yielded null findings for healthy food retail, and were similar to the main analysis except that the association of population density with CVD mortality became non-significant (online supplemental table S5). A sensitivity analysis at the census tract level was similar to the main analysis; the fully adjusted HR for any supermarket or produce market with cardiovascular mortality was not statistically significant and the

CI excluded any meaningful protective association (HR: 1.03; 95% CI 1.00 to 1.07) (online supplemental table S6). Likewise, analyses of healthy food retail presence with cardiovascular mortality did not result in a statistically significant association within any of the 11 county-based strata considered (table 4), though we note that the strongest trend in the hypothesised direction was for the 47 000 adults in counties assigned to the Southern Rural stratum (HR: 0.74; 95% CI 0.53 to 1.02). When continuous density was used instead of presence, each SD of healthy food source density was associated with slightly higher cardiovascular mortality, with confidence limits that exclude any HR supportive of our hypothesised direction of association (HR: 1.02; 95% CI 1.00 to 1.04, CBDRB-FY21-CES004-025).

We considered alternative indicators of presence of food retail by type (including both healthy and unhealthy sources) and broader cardiorespiratory and all-cause mortality outcomes (table 5). These variations in exposure and outcome definition did not result in healthy food retail being associated with reduced mortality; however, presence of healthy or unhealthy food retail was both associated with higher all-cause mortality.

**Table 2**  Correlation matrix for contextual variables, N=2753000

| | MHI | Pop den | Walkable | Supermkt | Healthyv1 | Healthyv2 | Fast food | Unhealthyv1 | Unhealthyv2 |
|---|---|---|---|---|---|---|---|---|---|
| Median household income (MHI) | 1 | | | | | | | | |
| Population density | 0.20 | 1 | | | | | | | |
| Walkable destination density | 0.17 | 0.97 | 1 | | | | | | |
| Supermarket density | 0.13 | 0.83 | 0.85 | 1 | | | | | |
| Supermarket or produce market (Healthy v1) | 0.13 | 0.87 | 0.88 | 0.96 | 1 | | | | |
| Healthy v1 + natural, health or vitamin stores (Healthy v2) | 0.16 | 0.92 | 0.94 | 0.91 | 0.94 | 1 | | | |
| Fast food density | 0.13 | 0.93 | 0.96 | 0.86 | 0.88 | 0.93 | 1 | | |
| Fast food, quick service, pizza, convenience, small grocery, bakery, coffee shop, candy, or ice cream (Unhealthy v1) | 0.14 | 0.97 | 0.99 | 0.85 | 0.88 | 0.94 | 0.97 | 1 | |
| Unhealthy v1 + nut stores, pharmacies, gas stations (Unhealthy v2) | 0.14 | 0.97 | 0.99 | 0.86 | 0.89 | 0.94 | 0.97 | 1.00* | 1 |
| | MHI | Pop Den | Walkable | Supermkt | Healthyv1 | Healthyv2 | Fast Food | Unhealthyv1 | Unhealthyv2 |

Values shown are Spearman rank correlation coefficients based on ZIP code tabulation area –based characteristics appended to individual-level records, all statistically significant with p<0.0001; CBDRB-FY20-022.
*Rounded from 0.998.
MHI, median household income.

## Discussion

While healthy food retail availability within the residential postal code area was hypothesised to be cardioprotective, we did not find support for this hypothesis in this large dataset representative of the continental USA. Findings were null (or in the opposite of the hypothesised direction where statistically significant) across tiered adjustment strategies, geographic units (ZCTA or census tract), across county-based strata defined using sociodemographic and climate data, and when clustering by county was accounted for using frailty models. In our exploration of other food retail variables and outcome specifications, presence of unhealthy food retail availability was noted to be associated with higher all-cause mortality.

Our overall finding that presence of healthy food retail was not associated with cardiovascular mortality echoes a recent finding that the association of food deserts with cardiovascular outcomes may predominately reflect associations with low area-based income rather than healthy food access.[8] The national scope of the present work leaves open the possibility that our classification is not sensitive to local variation in offerings across food venues or that features associated with healthy food retail presence (including unhealthy food sources) are obscuring a true causal association. The administrative geographic areas used for measuring the food environment are systematically larger in areas with low population density, yet may not fully reflect typical distance travelled for food acquisition[17] or optimise the correspondence with subjective experience and proximal behavioural outcomes.[24] However, recent reviews have questioned the strength of evidence linking geographically determined food environment measures to obesity,[25 26] relevant to the present work because obesity is a proposed mediator between the food environment and cardiovascular health. Gamba *et al*[26] note the highest proportion of significant findings in the expected direction among studies examining presence of food stores (vs proximity or density), the approach we have used; however, significant findings were noted to be commonly quite small and of borderline significance. Likewise, Cobb *et al*[25] conclude that findings to date on food environment and obesity are predominately null and raise concerns about quality and consistency. Qualitative findings relevant to the food environment and food behaviours have also been reviewed, with Pitt *et al*[27] noting salience in US contexts of food quality and affordability that varies among stores in a given category, as well as coping strategies that may importantly buffer effects of local food environment on behaviour. Limitations of GIS-based measures alone, without complementary information on pricing and shopper experience, are likewise underscored in a review of the food environment by Caspi *et al*.[28]

Nonetheless, further refinement of food environment exposure measures and investigation of associated cardiovascular morbidity and mortality may be warranted. Our analyses restricted to county-based strata across the USA (table 4) suggest such further investigation may

**Table 3** HRs and 95% CIs for association of healthy food retail with cardiovascular mortality, N=2 753 000 adults

| | Minimally adjusted | Moderate adjustment | Fully adjusted |
|---|---|---|---|
| Any supermarket or produce market present | 0.98 (0.95 to 1.02) | 1.03 (1.00 to 1.06) | 1.03 (1.00 to 1.07) |
| Female gender | **0.45 (0.44 to 0.46)** | **0.43 (0.42 to 0.44)** | **0.43 (0.42 to 0.44)** |
| Age (rescaled to per 10 years) | **2.72 (2.69 to 2.74)** | **2.64 (2.62 to 2.66)** | **2.64 (2.62 to 2.66)** |
| Married | **0.58 (0.57 to 0.59)** | **0.63 (0.61 to 0.64)** | **0.63 (0.62 to 0.64)** |
| US born | **1.35 (1.30 to 1.40)** | **1.30 (1.25 to 1.35)** | **1.31 (1.26 to 1.36)** |
| Black race | **1.08 (1.05 to 1.12)** | 1.00 (0.97 to 1.04) | **0.94 (0.91 to 0.98)** |
| Hispanic ethnicity | **0.89 (0.85 to 0.93)** | **0.80 (0.77 to 0.84)** | **0.76 (0.73 to 0.80)** |
| Educational attainment college or more | | **0.65 (0.63 to 0.67)** | **0.66 (0.64 to 0.68)** |
| Income (rescaled to per 10K) | | **0.97 (0.97 to 0.98)** | **0.98 (0.98 to 0.98)** |
| Median household income (rescaled to per 10K) | | | **0.96 (0.96 to 0.97)** |
| Population density (residents/km$^2$) (rescaled to per 10 K/km$^2$) | (shaded indicates exclude from model for the corresponding column) | | **1.12 (1.07 to 1.17)** |
| Walkable destination density (count/km$^2$), (rescaled to per SD) | | | 1.00 (0.98 to 1.01) |

Values show in each cell are HRs and 95% CIs from models with N=2 753 000; Boldface indicates statistical significance (p<0.05); CBDRB-FY20-CES004-030.

particularly be warranted in settings across the rural southern counties. Prior reviews and workshops support the salience of food environment for obesity and CVD prevention in such settings.[29 30]

While our a priori focus was on presence of healthy food retail and cardiovascular mortality, in analyses exploring alternative exposure and outcome specification we note that all food retail measures considered were associated with higher all-cause mortality. This was especially apparent for our most inclusive definition of unhealthy food sources. The presence of fast food or other venues promoting unhealthy eating may increase risk of cardiovascular mortality, as suggested by a large study in Canada.[31] In the last three decades, there has been an expansion of fast food outlets in the USA,[32 33] and an increased number of fast food restaurants in residential neighbourhoods has been investigated as a determinant of CVD outcomes and risk factors such as obesity.[1 34] Unhealthy food sources have the potential to increase consumption of highly processed and calorie dense foods.[13 35–38] Indeed, our results suggest unhealthy food store presence is associated with higher all-cause mortality.

**Table 4** Variation of across county strata for association of healthy food retail with cardiovascular mortality, N=2 753 000 adults

| Stratum | N | Minimally adjusted | Moderate adjustment | Fully adjusted |
|---|---|---|---|---|
| Southern Rural America | 47 000 | 0.74 (0.53 to 1.03) | 0.75 (0.54 to 1.04) | 0.74 (0.53 to 1.02) |
| North Central America | 112 000 | 1.03 (0.89 to 1.19) | 1.08 (0.93 to 1.25) | 1.10 (0.94 to 1.27) |
| Mid-Sized America | 127 000 | 0.92 (0.78 to 1.09) | 0.99 (0.84 to 1.16) | 0.97 (0.82 to 1.15) |
| Sunbelt America | 132 000 | 0.97 (0.86 to 1.09) | 0.99 (0.88 to 1.12) | 0.94 (0.83 to 1.06) |
| Poor America | 138 000 | 1.04 (0.94 to 1.04) | 1.06 (0.97 to 1.17) | 1.06 (1.00 to 1.17) |
| Mountain West America | 172 000 | 1.02 (0.90 to 1.15) | 1.04 (0.92 to 1.18) | 1.0 (0.90 to 1.16) |
| Beach America | 211 000 | 0.95 (0.83 to 1.08) | 0.96 (0.84 to 1.10) | 0.95 (0.83 to 1.09) |
| Wealthy America | 265 000 | 0.97 (0.86 to 1.11) | 0.99 (0.87 to 1.13) | 0.98 (0.86 to 1.12) |
| Middle America | 322 000 | 1.03 (0.94 to 1.13) | 1.08 (0.98 to 1.18) | 1.04 (0.95 to 1.14) |
| Northern Tier America | 330 000 | 0.96 (0.89 to 1.05) | 0.99 (0.91 to 1.08) | 1.00 (0.92 to 1.09) |
| Big City America | 509 000 | 1.03 (0.91 to 1.16) | 1.02 (0.90 to 1.15) | 0.98 (0.87 to 1.11) |

Values show in each cell are HRs and 95% CIs from models adjusted for gender, age, marital status, nativity, Black race, Hispanic ethnicity, educational attainment, income, median household income, population density and walkable destination density; CBDRB-FY20-CES004-038.

**Table 5** Variation of association across alternate definitions of healthy food store availability and alternate mortality outcomes

|  | Cardiovascular (38 500 deaths) | Cardiometabolic (87 000 deaths) | All to cause (247 000 deaths) |
|---|---|---|---|
| Healthy food store definition | | | |
| Supermarket | 1.01 (0.99 to 1.04) | **1.03 (1.01 to 1.05)** | **1.04 (1.03 to 1.06)** |
| Supermarket or produce market | 1.03 (1.00 to 1.07) | **1.04 (1.02 to 1.06)** | **1.05 (1.04 to 1.06)** |
| Supermarket, produce market, natural/health/vitamin store | **1.06 (1.02 to 1.10)** | **1.06 (1.03 to 1.09)** | **1.07 (1.05 to 1.09)** |
| Unhealthy food store definition | | | |
| Fast food restaurants | 1.03 (0.99 to 1.07) | **1.06 (1.03 to 1.09)** | **1.07 (1.06 to 1.09)** |
| Unhealthy food sources, restricted | 1.07 (0.97 to 1.17) | **1.11 (1.04 to 1.18)** | **1.15 (1.11 to 1.20)** |
| Unhealthy food sources, unrestricted | 1.05 (0.94 to 1.17) | **1.09 (1.01 to 1.17)** | **1.17 (1.12 to 1.23)** |

Values show in each cell are HRs and 95% CIs from models adjusted for gender, age, marital status, nativity, Black race, Hispanic ethnicity, educational attainment, income, median household income, population density and walkable destination density; CBDRB-FY20-CES004-043, CBDRB-FY21-CES004-025.

A comment is warranted on the consistent association noted for income with cardiovascular mortality. Both household and area-based income had a small but statistically significant association with reduced cardiovascular mortality across analyses. This echoes longstanding findings of a socioeconomic gradient across preventable adverse health outcomes including cardiovascular mortality.[39] When food desert measures defined jointly by both low-income and a lack of healthy food retail are associated with adverse health outcomes, the interpretation may falsely implicate the food environment and misdirect attention away from tackling more fundamental causes.

While caution should be taken in interpretation of covariate coefficients, given that our analysis strategy was not optimised with those coefficients in mind,[40] future work may be warranted to understand changes in the coefficient for Black racial identity from suggesting elevated risk in minimally adjusted models to a null or protective association following adjustment for socioeconomic and contextual characteristics. Attention is needed to structural racism and racial residential segregation[41] as well as continued discourse to counter any decontextualised biological interpretation of race.[42]

### Strengths and limitations
Strengths include the large, representative sample across the continental USA; individual, household, and area-level sociodemographic characteristics accounted for as potential confounders; and individual linkage to the NDI to examine cause-specific and all-cause mortality. Further, commercially licensed point-level retail data were cleaned and coded with attention to accuracy, consistency and transparency.[15] Finally, while main analyses were prespecified in the proposal process required for access to MDAC data, we incorporated sensitivity analyses to inform future research directions. In particular, since prior reviews have suggested effect modification by regional and population characteristics,[28] we incorporated stratified analyses and noted robustness of our null findings across strata.

However, several limitations should be noted. First, there may be uncontrolled confounding, as we did not have data on comorbidities and individual-level clinical or behavioural risk factors, which can be illustrated by the example of tobacco use. Cigarette smoking is potentially associated with area-based socioeconomic status, which in turn is associated with healthy food retail. We expect that controlling for individual and area-based socioeconomic status will minimise confounding by smoking, such that unmeasured confounding by smoking is unlikely to substantially account for the observed associations. However, these unmeasured characteristics could function as effect modifiers if, for example, medical advice while managing conditions such as diabetes alters how individuals respond to the local food environment.

Second, error likely remains in our linkage-based outcome assessment. Specifically, under-ascertainment of mortality among Hispanic and immigrant groups may result from return to country of origin at end of life or insufficient personal identifying data for unique linkage.[43]

Third, exposure mismeasurement may arise due to differences in duration of residence prior to 2008 or residential mobility during follow-up, which is not accounted for in our assessment of food retail and other independent variables. Further, our GIS-based assessment of the food environment relied on categories of retail, without complementary measures such as food pricing. A challenge we noted was the simultaneous consideration of multiple correlated density variables.

Finally, despite attempts to leverage a sampling strategy and corresponding weights to approximate a study population representative of US adults, there may be selection bias. This could have arisen at multiple points, including when respondents are given the option to decline permission for their data to be used for future research. While mean household income among our study sample is higher than the corresponding area-based median household income, suggesting that higher-income households may be over-represented, the contrast may reflect the

relative insensitivity of the median to inclusion of a small number of extreme high values typical of the skewed US income distribution.

## CONCLUSION

The hypothesised association of healthy food outlet presence (based on the residential postal code area) with reduced cardiovascular mortality was not supported in this nationally representative mortality follow-up study. This suggests that strategies aimed at addressing food deserts will miss opportunities for cardiovascular mortality improvement if the focus is exclusively on healthy food retail rather than addressing more foundational causes such as area-based income and opportunity.

**Contributors** The proposal, table planning, manuscript draft, and integration of coauthor comments were led by GL. Analyses were conducted by NJJ, who along with SA provided expert input into the appropriate use of and description of MDAC data. Input on methods, interpretation, and checking of table accuracy were provided by JRB. Longitudinal geographic characteristics were constructed and coded with expert input on the food retail classification (JH, KAM); potential built and social environment confounders (AR, KN); geospatial methods (JQ); and cardiovascular epidemiology (DS). All authors critically reviewed and approved of the manuscript prior to submission.

**Funding** This work was supported by the National Institute of Aging (grants 1R01AG049970, 3R01AG049970- 04S1), Commonwealth Universal Research Enhancement (C.U.R.E) program funded by the Pennsylvania Department of Health (2015 Formula award - SAP #4100072543). MDAC is supported by interagency agreements of both the National Institute on Aging and the National Heart, Lung, and Blood Institute with the U.S. Census Bureau. We also thank the Urban Health Collaborative at Drexel University, the Built Environment and Health Research Group at Columbia University, the Census Bureau, the Centers for Disease Control and Prevention and the National Institutes of Health for support in bringing together the data used in this research.

**Disclaimer** This paper is released to inform interested parties of research and to encourage discussion. Any views expressed on statistical, methodological, technical, or operational issues are those of the authors and not necessarily those of the U.S. Census Bureau. These results have been reviewed by the Census Bureau's Disclosure Review Board (DRB) to ensure that no confidential information is disclosed. The DRB release numbers are: CBDRB-FY20-CES004-013, CBDRB-FY20-CES004-021, CBDRB-FY20-022, CBDRBFY20-CES004-030, CBDRB-FY20-CES004-031, CBDRB-FY20-CES004-033, CBDRB-FY20-CES004-043, CBDRB-FY20-CES004-038, CBDRB-FY21-CES004-020. The views expressed in this manuscript are those of the authors and do not necessarily represent the views of the National Heart, Lung, and Blood Institute; the National Institutes of Health; or the U.S. Department of Health and Human Services.

**Competing interests** None declared.

**Patient consent for publication** Not required.

**Ethics approval** Ethical oversight of the research involvement of Drexel investigators was provided by the Human Research Protection Program in the Office of Research & Innovation at Drexel University (IRB Protocol: 1612004989). The Mortality Disparities in American Communities consists of responses for the full year 2008 American Community Survey (ACS) followed by over seven years of mortality tracking. The ACS survey data are collected under privacy and confidentiality provisions of the U.S. Census Bureau (Title 13, US Federal Code). The assurance of confidentiality of Census Bureau data is provided by Title 13 of the United States Code. As such, MDAC operational procedures carefully follow the well-defined practices designed to maintain the confidentiality of personal records as required by Title 13. These practices include the prevention of disclosure through the elimination of sparse cells in publications, the prohibited release of small-area geographical information on the MDAC public-use files, the use of an individually assigned MDAC control number to identify records instead of the use of personal identifiers for these purposes, and the restriction of persons having direct access to the MDAC database. In circumstances where MDAC participants requested restrictions on the use of their data by outside investigators, their information was not linked to mortality data.

**Provenance and peer review** Not commissioned; externally peer reviewed.

**Data availability statement** Data may be obtained from a third party and are not publicly available. Data sharing is restricted based on (1) terms of the licensing agreements for commercial establishment data and (2) screening of publicly released data or reports by the Census Bureau's Disclosure Review Board (CBDRB). Researchers interested to use the MDAC data can request access using a proposal-based process, described at https://www.census.gov/topics/research/mdac.html

**ORCID iDs**
Gina S Lovasi http://orcid.org/0000-0003-2613-9599
Janene R Brown http://orcid.org/0000-0003-3878-7999

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
