## [Reviewer comments · BMJ Open]

ARTICLE DETAILS

TITLE (PROVISIONAL)	Healthy Food Retail Availability and Cardiovascular Mortality Using Linked Data across the Contiguous US from the Mortality Disparities in American Communities Study
AUTHORS	Lovasi, Gina; Johnson, Norman; Altekruise, Sean; Hirsch, Jana; Moore, Kari; Brown, Janene; Rundle, Andrew; Quinn, James; Neckerman, Kathryn; Siscovick, David

VERSION 1 – REVIEW

REVIEWER	Claire Welsh Newcastle University, Population & Health Sciences Institute
REVIEW RETURNED	05-Feb-2021

GENERAL COMMENTS	Thank you for the opportunity to review this impressive manuscript which describes a large study of the association between provision of healthy food outlets and cardiovascular mortality in the US. I have no major concerns with the paper, and the authors are to be commended on their clear-sighted approach to the analysis and thoroughness of the discussion. I have a small number of comments or clarifications that they may, however, wish to address. 1. No mention is made, beyond the inclusion of an exposure to food outlets within walking distance, of how far from home people usually travel to buy food. Despite provision of healthy foods within a ZCTA, does this really equate to being near enough to people's homes that they would likely use such places? It may well be beyond the scope of this study, but identifying the average distance from the home of food outlets and comparing whether the distance is within peoples' 'usual' travel distance for food shopping may be of use. This is of particular note when assuming that the count of supermarkets, for example, is the same in two ZCTAs but due to differences in population density, this potentially means much greater travel distances to such places for those in the lower density area. If this cannot be done, then perhaps this needs to be addressed in the discussion. Similarly access to transport like cars could affect the likelihood of use of more distant healthy outlets.2. Supermarkets and small grocery stores are stockists of a variety of food types, both 'healthy' and 'unhealthy', so without knowing what is being purchased, there may be misclassification bias in these analyses. Please address this in the discussion, or add a sensitivity analysis where the 'mixed' outlets are omitted.3. Although adjustment for age, sex and socioeconomic status were included, you mention that it was not possible to exclude people with pre-existing cardiovascular morbidity. I think a little more discussion of this in the limitations is warranted, as it means that people who have been advised, due to such morbidity, to eat a 'healthier' diet, may be more likely to use healthier food outlets. If the proportion of
--

	this group varies by area it could affect results? 4. No data is reported (and is possibly unavailable) to describe both how long the food outlets have been present in those areas, and how long people have lived in their current homes, thus 'exposure' (opportunity to use them over time) to these places may vary greatly. Please mention this in the discussion. 5. I understand that mutual control for presence of healthy or unhealthy places was not possible, but I feel that models should have been adjusted for overall food outlet density. If this was not possible, please explain why and add to the limitations to explain how this omission could have affected the results. 6. Can you comment on why population density was not associated with cardiovascular mortality in men, but it was in women? 7. Were competing risk models considered to account for non-cardiovascular mortality? It may be worth running these as a sensitivity analysis. A few more specific points: Page 9, line 45: Were the implicit assumptions of the Cox PH model tested? How? What was the result? Also, what interactions were tested for? Tables: For those showing model results, please include a list of the covariates used in each adjustment model to avoid the reader having to go back through the text.
--	---

REVIEWER	Irene Dégano Hospital del Mar Institute for Medical Research
REVIEW RETURNED	17-Feb-2021

GENERAL COMMENTS	In the article "Healthy food retail availability is not associated with cardiovascular mortality in a representative US Sample" authors analyzed the association between healthy and unhealthy food availability and mortality in more than 2 million US citizens. The study and analyses are properly defined, results are clear, and the discussion compares and contrast many related issues. While the study is interesting for public health strategies some major points should be taken into account: Methods  - Why does the follow-up end in December of 2015? It has been more than 5 years since that date. Could the follow-up be extended? - Classification of healthy and unhealthy food retail is probably one of the key factors in this study. Supermarkets sell also unhealthy foods while bakery sell also healthy products. I would like to know how are convenience and grocery stores defined. In addition, I was wondering if you have run any sensitivity analysis with healthy food retail defined as produce stores alone. - Outcome definition could also be playing a role in the observed results. Why death from ischemic cerebrovascular disease is not included in the main cardiovascular mortality outcome? On the other hand, have the authors access to non-fatal cardiovascular disease events? It would be interesting to look at the association with all cardiovascular events. - Urban/non-urban classification is not clear to me. Was urban defined if in UA/UC and non-urban otherwise? - Have competing risks been taken into account in the analysis? Discussion  - The lack of association observed could be real or could be due to exposure/outcome misclassification and uncontrolled confounding among others. Regarding uncontrolled confounders authors name behavioral risk factors. Probably they should state diet and physical
--

	activity in addition to smoking. Individuals could have access to healthy foods but not buy them. Moreover, there are other community factors that could be playing a role in this study such as availability of healthy lifestyle education and availability of public physical activity facilities.  - The authors do not describe articles showing opposite results. Which are the differences between these studies and this one? - Potential exposure/outcome misclassification should be cited as a limitation.
--	--

REVIEWER	Shameer Khader Icahn Institute for Genomics and Multiscale Biology
REVIEW RETURNED	27-Feb-2021

GENERAL COMMENTS	Lovasi et al. address a critical public health question using a unique data resource. While the data set is large, the statistical validity and the assumptions made are questionable. I recommend the following queries/suggestions:  - Revise abstract. Add different strata information in the abstract - also share the different covariates and model adjustments - Define health vs. unhealthy food and its cost implications; for example, is healthy food more costly and not affordable? - I don't think simple association testing is enough to address these problems. For example presence of a supermarket with healthy but pricy options is as good as access to no healthy foods. I am wondering how authors could control/adjust for such confounders? - What's the distance threshold used to test the hypothesis? This should be defined in the abstract. - It is difficult to reproduce these results without access to the data or code related to the work; authors must provide a version of the data/code after applying suitable privacy measures to ensure reproducibility - ZSTAs have several challenges in their definition and assignment of locations. They are not an ideal indexing location to define the observations tested in the given study. An ideal approach would be to define a pre-defined zone around food sources and then look for health indices correlations. - Authors must correct the hypothesis from "Healthy Food Retail Availability Is Not Associated with Cardiovascular Mortality in a Representative US Sample" to "Healthy Food Retail Availability Is Not Associated with Cardiovascular Mortality in a Representative US Sample defined using ZCTA" - authors should correct this throughout the paper including the title - A section on the limitations of the study should be addressed and discuss various drawbacks of the study with representative examples - authors should provide a workflow of the study and inclusion/exclusion criteria - Authors must provide an appropriate EQUATOR statement suitable for the study - Authors must provide background on the data/code restriction and address it also as a limitation authors should provide-Information about IRB/consent from the census participants to opt-in/opt-out on such studies
---

VERSION 1 – AUTHOR RESPONSE

Reviewer 1

Thank you for the opportunity to review this impressive manuscript which describes a large study of the association between provision of healthy food outlets and cardiovascular mortality in the US. I have no major concerns with the paper, and the authors are to be commended on their clear-sighted approach to the analysis and thoroughness of the discussion. I have a small number of comments or clarifications that they may, however, wish to address.

Thank you for these kind words and for your thoughtful suggestions below.

R1.1. No mention is made, beyond the inclusion of an exposure to food outlets within walking distance, of how far from home people usually travel to buy food. Despite provision of healthy foods within a ZCTA, does this really equate to being near enough to people's homes that they would likely use such places? It may well be beyond the scope of this study, but identifying the average distance from the home of food outlets and comparing whether the distance is within peoples' 'usual' travel distance for food shopping may be of use. This is of particular note when assuming that the count of supermarkets, for example, is the same in two ZCTAs but due to differences in population density, this potentially means much greater travel distances to such places for those in the lower density area. If this cannot be done, then perhaps this needs to be addressed in the discussion. Similarly access to transport like cars could affect the likelihood of use of more distant healthy outlets.

This is an important topic for the field, and one which our research team and colleagues have found challenging. In other work, we have had more flexibility in defining geographic areas of interest, however as we note below there are legal and licensing restrictions for use of MDAC data and NETS data. As such, we address this important issue through modifications to the discussion section that can inform future work.

Flexibility for spatial scale was limited for the linked data used in this manuscript. Work with the individual-level MDAC data is subject to Title 13 restrictions, and our sharing of commercially licensed retail data (NETS data) are subject to terms set by the vendor, Walls and Associates. As such, we came to the feasible compromise of using reports summarizing retail availability for census tracts and ZIP code tabulation areas (ZCTAs) throughout the contiguous US, which could then be used at Census Headquarters to characterize the residential area for each MDAC participant meeting our inclusion criteria. In follow-up to a workshop earlier this year (<https://www.census.gov/library/video/2021/census-bureau-mortality-studies-workshop-access-to-data.html>) we continue to discuss with the MDAC Principal Investigators options to collaborate on more flexible incorporation of retail and other built environment data.

In planning to incorporate this important issue into the discussion, we note several competing ways to decide on the best geographic area to use for operationalizing the availability of food retail or other amenities:

- a) Use a fixed area definition guided by likely mode choice in accessing amenities.
- b) Align measurement with how participants delineate their own neighborhood.
- c) Extend areas to include the typical travel distance to access amenities of a certain type (such that areas characterized by longer distance traveled are larger).
- d) Select an area definition to maximize the correspondence between self-reported neighborhood measures and corresponding GIS-based metrics.
- e) Select an area definition to maximize the association between GIS-based metrics and proximal behavioral outcomes related to amenity use.

We point to this array to put the reviewer's suggestion (which corresponds most closely to c) in context. Each of these has some advantages and challenges. The approach we have taken is to use a fixed area definition (a, though also constrained as noted above), selecting ZCTAs in the proposal development stage as more likely to capture an area accessible by private vehicle. ZCTAs (and census tracts, used in our planned sensitivity analysis) have the characteristic of being larger in low population density areas (thus there may be some similarity to an ideal implementation of c). That ZCTAs are smaller in high density areas may be advantageous in approximating d and e (though that remains to be empirically tested), in so far as traffic congestion increases travel time for a given distance in high density areas, and alternative travel modes such as walking are more likely to be considered.

Alternatives that make participants' description or perception of their neighborhood central (b and d) have the advantage of recognizing heterogeneity in individuals' subjective experience of amenities and boundaries within their surroundings. Likewise, extending the areas used to include typical travel distances (c) points to the costs that some populations bear in traveling farther to access the nearest available, affordable, and acceptable amenities. A challenge is to consider whether the presence of an amenity such as a supermarket would itself change the neighborhood delineation, scale that best corresponds to perception, or travel distance. One way of viewing the longer travel distances to shop (typical of lower density areas) is that this reflects a lack of closer options (thus, the typical travel distance is itself capturing part of the exposure of interest, as we consider amenity availability). In fact, if we personalized neighborhood definitions to always include the usual shopping venue, we would have eliminated much of the relevant exposure variation. Of course, another way of viewing this is that the broader geographic context may alter whether any given geographic area definition is appropriate, and with that in mind we have included sensitivity analyses restricting analyses to urban areas, or to county-based strata based on contextual characteristics. Ultimately, we posit that comparing geographic areas based on predictive validity of corresponding built environment metrics (e) may provide the needed empirical guidance for future research, especially if used alongside qualitative perspectives and activity space data captured using GPS. (See Hirsch, J. A., & Hillier, A. (2013). Exploring the role of the food environment on food shopping patterns in Philadelphia, PA, USA: a semiquantitative comparison of two matched neighborhood groups. *International journal of environmental research and public health*, 10(1), 295-313.)

In the discussion, our comments on this issue are brief due to space constraints. Importantly we were able to direct readers to a key reference by Michele Ver Ploeg and colleagues and suggest considerations for future work.

“The administrative geographic areas used for measuring the food environment are systematically larger in areas with low population density, yet may not fully reflect typical distance traveled for food acquisition¹⁷ or optimize the correspondence with subjective experience and proximal behavioral outcomes.²⁴”

R1.2. Supermarkets and small grocery stores are stockists of a variety of food types, both ‘healthy’ and ‘unhealthy’, so without knowing what is being purchased, there may be misclassification bias in these analyses. Please address this in the discussion, or add a sensitivity analysis where the ‘mixed’ outlets are omitted.

Thank you for bringing forward another important topic, which our team discussed often in our efforts to categorize establishments for health research purposes. In our classification of retail as providing healthy or unhealthy food, we considered prior literature as well as

- a) what items are available
- b) the relative affordability of fresh items, especially produce
- c) what we understood to be the usual intent of customers

As the reviewer points out, there are food stores and restaurants where classification according to the availability of items (a) alone would be indeterminate. In particular, supermarkets stock highly processed foods as well as healthier items such as fresh produce; fast food restaurants include healthy items, such as salads, on their menu alongside calorie dense items with less nutritious makeup. However, prior literature plus consideration of pricing and purchasing patterns suggest that supermarkets are an important source of affordable fresh items and are likely visited with the intent of at home meal preparation meeting customer goals which may include health. To the methods section, we have added the following sentence:

“While we recognize that establishments within the above categories offer items with varying nutritional value, our categorization was informed by prior literature and by the relative affordability of and salience of fresh items.”

We recognize that our classification system has simplified as dichotomous what may be best considered as a continuous spectrum; we do examine variations in exposure definition to provide some insight into how sensitive our results are to this (Table 5). In addition, we mention in the discussion that there may be important local variation missed by our approach.

“The national scope of the present work leaves open the possibility that our classification is not sensitive to local variation in offerings across food venues or that features associated with healthy food retail presence (including unhealthy food sources) are obscuring a true causal association.”

R1.3. Although adjustment for age, sex and socioeconomic status were included, you mention that it was not possible to exclude people with pre-existing cardiovascular morbidity. I think a little more discussion of this in the limitations is warranted, as it means that people who have been advised, due to such morbidity, to eat a ‘healthier’ diet, may be more likely to use healthier food outlets. If the proportion of this group varies by area it could affect results?

We discuss the potential for unmeasured confounding briefly in our limitations section, using smoking status to illustrate that adjustment for area-based socioeconomic status may be sufficient to block some pathways involving unmeasured potential confounders. The reviewer seems to suggest also a potential for effect modification of medical indication to change diet, which is interesting to consider for other work. We modified our discussion to include this possibility as follows:

“We expect that controlling for individual and area-based socioeconomic status will minimize confounding by smoking, such that unmeasured confounding by smoking is unlikely to substantially account for the observed associations. However, these unmeasured characteristics could function as effect modifiers if, for example, medical advice while managing conditions such as diabetes alters how individuals respond to the local food environment.”

R1.4. No data is reported (and is possibly unavailable) to describe both how long the food outlets have been present in those areas, and how long people have lived in their current homes, thus ‘exposure’ (opportunity to use them over time) to these places may vary greatly. Please mention this in the discussion.

The issues of changes to the food environment over time, and length of resident in current home are both important to consider.

Regarding changes to the food environment and other types of retail establishments, we do have the ability to make comparisons over time using the National Establishment Time Series data. Findings we have published for the trajectory of growth in unhealthy food sources in New York City (Berger, N., Kaufman, T. K., Bader, M. D., Rundle, A. G., Mooney, S. J., Neckerman, K. M., & Lovasi, G. S. (2019). Disparities in trajectories of changes in the unhealthy food environment in New York City: a latent class growth analysis, 1990–2010. *Social Science & Medicine*, 234, 112362) suggest that despite growth in unhealthy food sources over time, census tracts tend to remain relatively high or relatively low in unhealthy food source density (that is, the most rapid growth was in the places with highest density of unhealthy food source establishments at baseline). Building on these findings, we have noted that growth in unhealthy food sources has been more rapid than other food source categories nationally. However, as we want to provide appropriate context into how the patterns we note vary spatially and fit in with the prior literature on access disparities, this is being developed as a separate paper (an abstract for this work has been presented by Jana Hirsch (a coauthor on this paper) at the Interdisciplinary Association for Population Health Science conference).

Regarding duration of residence at the present address, information in the American Communities Survey is limited. There are questions about residential address for each household member one year prior to the survey, and the year that survey respondent moved into the address reported in 2008. We had mentioned in the discussion section the possibility of moving to a new address between 2008 and 2015, and modified this statement to also acknowledge the effect of residential duration prior to 2008:

“Third, exposure mismeasurement may arise due to duration of residence prior to 2008 or residential mobility during follow-up, which is not accounted for in our assessment of food retail and other independent variables.”

R1.5. I understand that mutual control for presence of healthy or unhealthy places was not possible, but I feel that models should have been adjusted for overall food outlet density. If this was not possible, please explain why and add to the limitations to explain how this omission could have affected the results.

Distinguishing the health effects across multiple correlated measures of the food environment is an important challenge for the field. Mutual adjustment and use of ratio measures (e.g., proportion of available food sources classified as unhealthy) both have been used, as have food desert measures that combine food environment and socioeconomic context.

Our research question in this case centered around the presence of healthy food sources (primarily defined as supermarket or produce market presence), which is common but not universal at the ZCTA level. Unhealthy food sources and walkable destinations are both concentrated in the ZCTAs with healthy food presence. As shown in the correlation matrix (Table 2), the density or walkable destinations was highly correlated with our largest food source category (unhealthy, version 2). By controlling for walkable destination density in our fully adjusted models, we endeavor to distinguish the food environment effects from other pathways featuring broader health effect of commercial density (that could be mediated by economic opportunity, stress, or physical activity pathways). There were minimal differences between fully adjusted models in comparison to moderately adjusted models with only demographic and socioeconomic adjustment (see Table 3, as well as supplemental tables).

Thus, while adjustment for total food outlets is possible, we argue that adjustment for walkable destination density has served the same intended purpose, while having the attractive feature of being separate from the food environment pathway of interest.

R1.6. Can you comment on why population density was not associated with cardiovascular mortality in men, but it was in women?

This is an interesting observation, and we thank the reviewer for the close inspection of supplemental materials. Interpretation of coefficients for covariates is challenging, as we note in the discussion.

“While caution should be taken in interpretation of covariate coefficients, given that our analysis strategy was not optimized with those coefficients in mind,³⁷ future work may be warranted...”

Another MDAC proposal is currently in progress that focuses on population density and other indicators of walkability that may relate to travel mode choice (with some overlap in the writing team, and having been presented by Alex Quistberg at the Interdisciplinary Association for Population Health Science conference). Given efforts to respond to comments and requests for clarification in the present paper while keeping the work concise, we propose to defer discussion of effect modification by sex for population density to future work in which density is itself incorporated into the exposure of interest.

R1.7. Were competing risk models considered to account for non-cardiovascular mortality? It may be worth running these as a sensitivity analysis.

Thank you for this suggestion. In our work, we treated death from other causes as censoring events in our main analysis (cause-specific hazard models), and then also examined models of all-cause mortality (thus, other deaths considered as failures).

Using Fine and Gray’s approach to obtain subdistribution hazard ratios would be valuable as a future research direction, allowing comparison of the direction of association across other chronic disease causes of death that may be similarly responsive to variation in the food environment (such as obesity-associated cancers) or causes which provide a contrast relevant to ruling out competing explanations such as confounding (such as traffic-related injuries that are not expected to be diet related). We did run competing risk models in SAS and also replicated them in stata for our minimally adjusted main model, and we found consistency in the direction and statistical significance of the observed associations.

However, as our approved proposal and a priori plan was to focus on cardiovascular causes, we argue that competing risk models do not provide substantial added value and have not modified our manuscript to include them. Because MDAC only includes fatal events, the event of interest and competing risk (cardiovascular death or death from other causes) cannot be both observed in the same individual. Further, the following resource, which was found to be helpful when working on this revision, suggests that cause-specific hazard models may be preferred for etiologic work such as ours:

Austin, P. C., & Fine, J. P. (2017). Practical recommendations for reporting Fine-Gray model analyses for competing risk data. *Statistics in Medicine*, 36(27), 4391-4400.

A few more specific points:

R1.8. Page 9, line 45: Were the implicit assumptions of the Cox PH model tested? How? What was the result? Also, what interactions were tested for?

Thank you for bringing attention to model diagnostics and tested interactions. We used proc lifetest to evaluate the proportional hazards assumption for our Cox PH models, and noted no violation of the proportional hazards assumption for our exposure of interest (for the minimally adjusted model $p = 0.45$, for the moderately adjusted model $p = 0.72$, and for the fully adjusted model $p = 0.91$). Of potential interest for future work, there were however suggestions that proportional hazards may be violated for commonly used covariates including male sex and Black race ($p < 0.001$ and $p = 0.04$, respectively, in minimally adjusted models).

R1.9. Tables: For those showing model results, please include a list of the covariates used in each adjustment model to avoid the reader having to go back through the text.

Thank you for this suggestion. Footnotes have been modified to list covariates for Tables 4-5 and Figure S1.

Reviewer: 2

In the article “Healthy food retail availability is not associated with cardiovascular mortality in a representative US Sample” authors analyzed the association between healthy and unhealthy food availability and mortality in more than 2 million US citizens. The study and analyses are properly defined, results are clear, and the discussion compares and contrast many related issues. While the study is interesting for public health strategies some major points should be taken into account:

Thank you for your interest in our work, and your attention to the important points that follow.

Methods

R2.1. Why does the follow-up end in December of 2015? It has been more than 5 years since that date. Could the follow-up be extended?

An update to the NDI linkage is planned for MDAC, but has not been completed as of this time. Updating our analyses to include more recent years of follow-up is not feasible within the time allotted for revision.

R2.2. Classification of healthy and unhealthy food retail is probably one of the key factors in this study. Supermarkets sell also unhealthy foods while bakery sell also healthy products. I would like to know how are convenience and grocery stores defined. In addition, I was

wondering if you have run any sensitivity analysis with healthy food retail defined as produce stores alone.

Please see our response to this issue, raised by comment R1.2 above. Regarding produce stores alone, this relatively small category could be considered in future work. In doing so, we would recommend that investigators interested in that specific amenity type consider mobile and time limited produce sources (e.g., carts and farmers markets) that may require the incorporation of data sources beyond the NETS data, which we have relied on to construct our food environment measures.

R2.3. Outcome definition could also be playing a role in the observed results. Why death from ischemic cerebrovascular disease is not included in the main cardiovascular mortality outcome? On the other hand, have the authors access to non-fatal cardiovascular disease events? It would be interesting to look at the association with all cardiovascular events.

Thank you for raising the outcome definitions, as your suggestions point to important future directions in this and other data sources.

Our primary and secondary outcome definitions were planned at the stage of developing an MDAC proposal. We used a standardized list of 113 selected causes of death from the Center for Disease Control and Prevention National Center for Health Statistics, because these were readily available for analyses and documented in the MDAC reference manual (Appendix G). In this list Cerebrovascular diseases (code: 61) are not distinguished as ischemic or not. Thus, we decided that we would start with an approach examining the sensitivity of results to outcome definitions that could be constructed using the 113 selected causes. We have updated the methods text to be clear that this list of causes is the basis for our outcome definition:

“The primary cardiovascular mortality outcome based on NDI (based on 113 selected causes of death as defined by the Center for Disease Control and Prevention National Center for Health Statistics) included...”

However, future work could use the ICD-10 codes to refine the outcome definition, as well as including up to three underlying causes.

Based on the nature of the MDAC data, we had no ability in this study to consider non-fatal events. Future use could focus on the subset of individuals for whom linkage to medical claims data is available. However, at the moment an alternative and promising strategy for consideration of non-fatal events is work within cardiovascular cohort studies, as some members of our writing team are doing, to complement this work.

R2.4. Urban/non-urban classification is not clear to me. Was urban defined if in UA/UC and non-urban otherwise?

Yes, urban was defined as a dichotomous variable for purposes of a restricted analyses presented in the online supplement. MDAC participants were classified based on their residence being within a UA or a UC (urban = 1) or not (urban = 0). We have rephrased in the methods section to clarify.

“Residential location of each MDAC household was classified as urban if located within an urbanized area (UAs) or urban cluster (UCs). Urbanized Areas (UAs) consist of densely developed territories that contain 50,000 or more people. Urban Clusters (UCs) consist of densely developed territories with at least 2,500 people but fewer than 50,000 people. In 2010, an estimated 81% of the US population resided in urban areas.²⁰”

R2.5. Have competing risks been taken into account in the analysis?

Please see response to R1.7 above for a discussion of our approach, and the potential role of competing risk analysis in future work.

Discussion

R2.6. The lack of association observed could be real or could be due to exposure/outcome misclassification and uncontrolled confounding among others. Regarding uncontrolled confounders authors name behavioral risk factors. Probably they should state diet and physical activity in addition to smoking. Individuals could have access to healthy foods but not buy them. Moreover, there are other community factors that could be playing a role in this study such as availability of healthy lifestyle education and availability of public physical activity facilities.

The reviewer is certainly correct that there could be multiple explanations for the observed findings. In view of its importance, the possibility of uncontrolled confounding is mentioned first among our limitations.

Smoking is used to illustrate that for behavioral risk factors that could be of concern, yet we are only in this instance able to mitigate (but not eliminate) the concern through adjustment for contextual characteristics. If smoking is a confounder, the distortion of our association of interest may be attenuated or blocked entirely through appropriate control for individual and area-based socioeconomic characteristics. Likewise for physical activity, confounding may be attenuated or blocked through control for contextual characteristics indicative of walkability (population density and walkable destination density).

However, regarding behavioral factors such as diet, we would consider that there may be a risk of over adjustment should we be in a position to treat diet as a confounder. Diet-associated neighborhood preferences would, however, be an attractive variable to pursue as a common prior cause influencing neighborhood selection and subsequent cardiovascular mortality. Nonetheless, mediation will be important to explore in future work, such that being undertaken with harmonized data across the Diabetes LEAD Network (as described in Hirsch, A. G., Carson, A. P., Lee, N. L., McAlexander, T., Mercado, C., Siegel, K., ... & Thorpe, L. E. (2020). The diabetes location, environmental attributes, and disparities network: Protocol for nested case control and cohort studies, rationale, and baseline characteristics. *JMIR research protocols*, 9(10), e21377).

R2.7. The authors do not describe articles showing opposite results. Which are the differences between these studies and this one?

In an effort to be concise, we relied on selective citation of prior original research, and refer readers to several relevant systematic reviews that take a more comprehensive view of the literature.

R2.8. Potential exposure/outcome misclassification should be cited as a limitation.

Among the limitations discussed, we have paragraphs dedicated to both outcome (starting with “Second, error likely remains in our linkage-based outcome assessment...”) and exposure misclassification (starting with “Third, exposure mismeasurement may arise...”). The language of the latter has been updated based on reviewer suggestion R1.5 above.

Reviewer: 3

Lovasi et al. address a critical public health question using a unique data resource. While the data set is large, the statistical validity and the assumptions made are questionable.

Thank you for bringing attention to both the importance of our research question and the opportunities to provide appropriate clarification and caveats in sharing this work.

R3.1. Revise abstract. Add different strata information in the abstract - also share the different covariates and model adjustments

Per editorial guidance (see response to comment E.2) and in light of these suggestions, we have substantially rewritten our abstract. Most relevant to this comment is the addition of parenthetical details to the following revised statement:

“Healthy food retail presence was not associated with reduced cardiovascular (HR: 1.02; 95% CI: 0.99-1.06) or all-cause mortality (HR: 1.04; 95% CI: 1.03-1.05) in fully adjusted models (with adjustment for gender, age, marital status, nativity, Black race, Hispanic ethnicity, educational attainment, income, median household income, population density, and walkable destination density), or in sensitivity analyses and strata (based on gender, urbanicity, household type, or county-based typology) considered.”

R3.2. Define health vs. unhealthy food and its cost implications; for example, is healthy food more costly and not affordable?

Please see response to R1.2 above, including the clarification that our classification does give some consideration to the relative affordability of fresh items by establishment type. In our discussion of prior literature, we do note the importance of affordability of foods in the US context:

“Qualitative findings relevant to the food environment and food behaviors have also been reviewed, with Pitt and colleagues²⁷ noting salience in US contexts of food quality and affordability that varies among stores in a given category, as well as coping strategies that may importantly buffer effects of local food environment on behavior.”

R3.3. I don't think simple association testing is enough to address these problems. For example presence of a supermarket with healthy but pricy options is as good as access to no healthy foods. I am wondering how authors could control/adjust for such confounders?

As noted in response to R1.2, we have used prior literature to inform our definitions, yet we acknowledge that this approach has limitations, as mentioned in the discussion:

“Further, our GIS-based assessment of the food environment relied on categories of retail, without complementary measures such as food pricing.”

R3.4. What's the distance threshold used to test the hypothesis? This should be defined in the abstract.

Our food environment exposures are defined primarily based on presence within the ZCTA, with sensitivity analyses considering instead the census tract. Reliance on such administrative areas was necessitated for these data sources, and the strengths and alternatives are discussed above in our response to comment R1.1.

R3.5. It is difficult to reproduce these results without access to the data or code related to the work; authors must provide a version of the data/code after applying suitable privacy measures to ensure reproducibility.

Data cannot be shared due to Title 13 and commercial licensing restrictions. Researchers interested to validate or build upon our findings can gain access to the MDAC data through submission of a proposal (see https://www.census.gov/topics/research/mdac.Research_Proposal.html). Some content of statistical syntax files is considered sensitive due to revealing file structures. However, to provide more transparency on our inclusion criteria, failure definition, and modeling approach, we provide in this letter key sections of the SAS syntax used:

Code for MDAC MS # M016

A. Code defining mortality indicators for primary CVD outcome (*indcvd*) and alternative expanded CVD outcome (*indcvd2*) for use as an indicator of failure in time to event analyses, along with corresponding categorical variables used for competing risk analyses (*cvdcomprisk*, *cvd2comprisk*):

```
indcvd = 0;

    if cause113 in('052' '053' '054' '055' '062') then indcvd = 1;

indcvd2 = 0;

    if cause113 in('052' '053' '054' '055' '062' '043' '050'
        '051' '058' '059' '060' '061' '062' '063'
        '064' '065') then indcvd2 = 1;

cvdcomprisk = 0;

    if indcvd = 1 then cvdcomprisk = 1;

    else if indcvd = 0 and matchstat = '2' then cvdcomprisk = 2 ;

cvd2comprisk = 0;

    if indcvd2 = 1 then cvd2comprisk = 1;

    else if indcvd2 = 0 and matchstat = '2' then cvd2comprisk = 2 ;
```

B. Code showing exclusions (based on restricting to a subset without missing covariate data [for gender, marital status, place of birth, race, ethnicity, educational attainment, household income], age

25+, and in households that are not group quarters and that have full kitchen facilities) applied to create the analytical dataset from the full MDAC analytical database:

```

data lovasmdacrecvd;

    set sourcefile ;

    if sex in ('1' '2') ; * RECORD HAS VALID GENDER CODE;

    if 25 <= age <= 300 ; * RECORD HAS VALID AGE;

    if '1' <= mar <= '5' ; * RECORD HAS VALID MARITAL STATUS CODE;

    if '01' <= pobr <= '10' ; * RECORD HAS VALID POB CODE;

    if '01' <= imprc <= '31' ; * RECORD HAS VALID RACE CODE;

    if '1' <= hsgp <= '8' ; * RECORD HAS VALID HISPANIC CODE;

    if '01' <= schl <= '24' ; * RECORD HAS VALID EDUCATION CODE;

    if hinc_cat in ('-$' '00' '01' '02' '03' '04' '05' '06' '07' '08' '09' '10' '11' '12') ;

    if gq = 0; * KEEP RECORDS NOT IN GROUP QUARTERS;

    if kit = '1' ; * KEEP RECORDS WITH COMPLETE KITCHEN
FACILITIES;

run;

```

C. Code to illustrate the Cox proportional hazards modeling approach, shown here for the minimally adjusted main analyses (exposure of interest is a dichotomous variable indicating whether supermarkets or produce markets were present in the ZCTA: *indanyhealthyfoodz*):

```

proc phreg data=lovasmdacrecvd fast;

    title1 '*****',

    title2 '* PROJECT:Availability of Health food Sources & CVD MDAC MS# 16 *';

    title3 '* Author: Prof. Gina Lovasi, Drexel University CVD Mortality *';

    title4 '* ----- MINIMALLY ADJUSTED MODEL ----- *';

    title5 '* ANALYSIS INVOLVED: Cox Model Regressions *';

    title6 '* DATA INVOLVED: MDAC & Drexel Food Services Availability Data *';

    title7 '* FILES INCLUDED: ALL RUN DATE:02/06/20*';

    title8 '**** TABLE 2c URBAN ONLY *** ZCTA ANALYSIS *** WEIGHTED DATA ****';

    model followdays*indcvd(0) =

```

```
indanyhealthyfoodz  
indfemale  
age10rescale  
indmarried  
indusborn  
indblack  
indhisp  
/ RL ;  
weight mdac_wgt;  
  
run;
```

R3.6. ZSTAs have several challenges in their definition and assignment of locations. They are not an ideal indexing location to define the observations tested in the given study. An ideal approach would be to define a pre-defined zone around food sources and then look for health indices correlations.

Please see discussion of related issues in response to R1.1. The ideal approach described may be explored in future work with more flexibility to have simultaneous access to point level residential and retail data.

R3.7. Authors must correct the hypothesis from "Healthy Food Retail Availability Is Not Associated with Cardiovascular Mortality in a Representative US Sample" to "Healthy Food Retail Availability Is Not Associated with Cardiovascular Mortality in a Representative US Sample defined using ZCTA" - authors should correct this throughout the paper including the title

The title has been changed with editorial guidance. While neither ZCTA-based nor census tract-based analyses supported our hypotheses, we reviewed the text and found opportunities within several sections to clarify that the findings are specific to the residential ZCTA, considered *a priori* as primary among the two administrative areas used. Thank you for this suggestion.

R3.8. A section on the limitations of the study should be addressed and discuss various drawbacks of the study with representative examples.

We do discuss limitations, though we have limited the space dedicated to illustrating limitations with examples in an effort to be concise.

R3.9. Authors should provide a workflow of the study and inclusion/exclusion criteria

We note that we have a dedicated section within the methods to describe the inclusion criteria (please see corresponding syntax in response to R3.5 above). As noted within this section, while we can illustrate the change in sample size as each inclusion criteria is applied, we do so only with numbers rounded to the thousands due to protections against disclosure of personal information that are required for use of this data source.

“Our analytic sample was initially restricted to individuals from ACS survey households with consent for research data use (N=4,512,000; note that sample sizes in tables and to illustrate changes as inclusion criteria are applied are rounded to the thousands during disclosure proofing; CBDRB-FY20-CES004-021). We further limited to individuals for whom personal identifiers were sufficiently complete to allow linkage to NDI through December 31, 2015 (4,480,000). Due to potential differences in food acquisition, we excluded individuals residing in group quarters or in households without a full kitchen (3.8%). Linkage to ZCTA-level food environment data assembled across the continental US was completed for 4,107,000 individuals. Based on our interest in associations with cardiovascular mortality adjusted for individual socioeconomic characteristics, we restricted our analyses to adults 25+ years of age (2,923,000). Final exclusion of observations with missing covariate data resulted in an analytic sample of 2,753,000.”

Regarding MDAC data, we note that the study website (https://www.census.gov/topics/research/mdac.Data_Availability.html) offers detailed documentation and video to help potential users become oriented to the data available. Regarding the NETS data, we have a prior publication dedicated to the methodology (Hirsch, Jana A., Kari A. Moore, Jesse Cahill, James Quinn, Yuzhe Zhao, Felicia J. Bayer, Andrew Rundle, and Gina S. Lovasi. "Business data categorization and refinement for application in longitudinal neighborhood health research: a methodology." *Journal of Urban Health* (2020): 1-14.), which includes a relevant diagram (Figure 1).

R3.10. Authors must provide an appropriate EQUATOR statement suitable for the study

We previously completed a STROBE cohort checklist to include with our submission.

R3.11. Authors must provide background on the data/code restriction and address it also as a limitation authors should provide-Information about IRB/consent from the census participants to opt-in/opt-out on such studies

The MDAC data is unusual in being subject to Title 13 restrictions (please see also our response to comment R3.5 above). Our analyses excluded data on any individuals who indicated refusal of research use of their data, and all results were subject to disclosure proofing prior to inclusion in the

manuscript or supplemental materials. The contributions to this work by Drexel personnel were covered by a protocol approved by Drexel University's Institutional Review Board.

VERSION 2 – REVIEW

REVIEWER	Claire Welsh Newcastle University, Population & Health Sciences Institute
REVIEW RETURNED	18-May-2021

GENERAL COMMENTS	I congratulate the authors on comprehensively addressing all reviewer comments. I have no further comments and look forward to seeing this manuscript published.
--

REVIEWER	Shameer Khader Icahn Institute for Genomics and Multiscale Biology
REVIEW RETURNED	26-May-2021

GENERAL COMMENTS	I would like to thank the authors for addressing all of my comments. I would recommend the editorial team consider accepting this manuscript for publication after a statistician/epidemiologist review if it has not been done until now.
--